# Harmonization of Epidemiologic Research Methods to Address the Environmental and Social Determinants of Urban Slum Health Challenges in Sub-Saharan Africa

**DOI:** 10.3390/ijerph191811273

**Published:** 2022-09-08

**Authors:** Adetoun Mustapha, A. Kofi Amegah, Eric Stephen Coker

**Affiliations:** 1Nigeria Institute of Medical Research, 6 Edmund Crescent, Lagos 101245, Nigeria; 2Department of Biomedical Sciences, School of Allied Health Sciences, University of Cape Coast, Cape Coast PMB TF0494, Ghana; 3Department of Environmental & Public Health, College of Public Health & Health Professions, University of Florida, Gainesville, FL 32610, USA

**Keywords:** urban slum, exposure assessment, research methods, socio-economic status, Sub-Saharan Africa, air pollution, health

## Abstract

Sub-Saharan Africa (SSA) has a significant proportion of populations living in urban slum conditions, where exposure to multiple environmental stressors and social inequalities is ubiquitous. This commentary synthesizes commonalities in recent environmental health studies from urban cities in East and West Africa, presented during a symposium sponsored by the Africa Chapter of the International Society of Environmental Epidemiology (ISEE) in August 2020. A key takeaway from this symposium is the need for harmonization of epidemiologic and exposure data collection in three domains tailored to the SSA context: (1) improvements in socioeconomic status (SES) measurement through harmonization in the conceptualization and operationalization of SES indicators; (2) improvements in air pollution exposure assessment in resource-constrained contexts by better integration, validation, and harmonization of exposure data of air pollution and mitigating factors; and (3) harmonization in the assessment of health outcomes and biomonitoring of contaminants. Focusing on these three domains would galvanize environmental epidemiologists in SSA around shared data collection instruments and shared data platforms and facilitate the pooling of data across the continent. Fostering this collaborative research will enable researchers and decision-makers to glean new insights and develop robust environmental health interventions and policies for SSA urban slums and for improved population health.

## 1. Overview of Environmental Exposures and Human Health Effects in Sub-Saharan Africa’s Urban Slums

Rapid urbanization without adequate planning or infrastructure capacity has contributed to the development of informal settlements or slums in low- and middle-income countries. Major cities in Sub-Saharan Africa (SSA) are home to some of the world’s largest slum settlements. The Kibera slum in Nairobi is the largest urban slum in Africa (Amegah, 2021) [1]. Sub-Saharan Africa records the highest percentage of slum dwellers, with about 54% of its urban population residing in slums in [2]. Slum settlements often have poor and damp housing, poor sanitation, open sewage, worsening air quality from unpaved roads, increased use of biomass fuels for cooking, and open burning of solid waste. These deplorable environmental conditions expose slum dwellers to pathogens and ill-health. Lack of pipe-borne water in slum areas has also meant the consumption of unsafe water owing to reliance on ground and surface water which become polluted by open drains and nearby pit latrines. Pit latrines are also in widespread use in these settlements. The formation and proliferation of slums in African cities are a critical manifestation of social exclusion, for which the socioeconomic and health impacts are not yet thoroughly studied [3]. Air pollution exposure in urban slums ranges from fine particulate matter, polycyclic aromatic hydrocarbons, and other gaseous pollutants such as nitrogen oxides. While exposure to heavy metals in Africa’s slums [4] is not well studied, in a biomonitoring study of 100 Ugandan children living in an urban slum of Kampala, 97% of the children sampled had elevated blood lead levels. Yet, researchers could not ascertain the sources of their lead exposures due to a lack of research in the SSA slum context [5].

These environmental exposures in slum settlements undoubtedly contribute to the high burden of disease and ill-health, including cholera, gastroenteritis, respiratory illnesses, adverse pregnancy outcomes and cardiovascular diseases [6,7,8,9]. Slums are also known for their atmosphere of insecurity and violence [10] and material and economic deprivation. Yet, there is virtually no research into the cumulative and interacting socioeconomic and environmental factors that are at play, at the household level, which create a complex web of social determinants of health in Africa’s urban settings.

## 2. Conceptualization and Operationalization of Socioeconomic Status (SES) in Africa’s Urban Context

The social determinants of health were recognized early on in Europe, with the publication of Rudolf Virchow’s “Report on the Typhus Epidemic in Upper Silesia” and Friedrich Engels’ “Condition of the Working Class in England” (1844) [11,12]. Hence, the role of SES as a key confounding and effect modifying variable in environmental epidemiological studies is well-recognized. However, it follows that the conceptualization and operationalization of SES in environmental epidemiological research also developed from the high-income country contexts of European and North American countries, where single and composite measures from census and administrative databases of local authorities have been used to define socioeconomic characteristics of an area. Like many other developing countries, the current situation in Sub-Saharan African countries is that demographic data at the district level is scarce. At the same time, the methodologies for defining socioeconomic groups are less developed. Given the vast differences between the higher income country urban context and SSA urban context, the conceptualization and operationalization of SES for epidemiologic research must be tailored to SSA’s contemporary urban context. In our ISEE symposium, we presented research from urban cities of Nigeria and Uganda. In the Ugandan study [9], SES was classified using a household assets index from data collected via a household questionnaire. Specifically, the household assets index was a composite summation score of multiple assets reported for each study household, including electricity, a television, a motor vehicle, a closet (‘wardrobe), a CD or DVD player, and a bicycle (range: 6–12; no = 1 and yes = 2 for each asset). An increasing assets index indicated an increasing number of household assets and thus higher SES for higher index scores. To generate the neighborhood SES in the Nigerian study [13], a purpose-designed measure of deprivation was devised, using data that could be gathered in the field. Neighborhood conditions were defined based on indicators that have been demonstrated to be determinants of health outcomes from literature [14,15,16,17,18] and material deprivation context that have been highlighted to play a role in health outcomes in Africa from global initiatives such as the WHO/UNICEF Global Water Supply and Sanitation Assessment (2000) [19]. Data relating to these indicators were gathered through a surveillance exercise. This involved driving and walking through the area, visiting schools, and interacting with residents of the area on three to five visits. The general characteristics of the neighborhood were observed and recorded during these exercises. The neighborhood SES was defined based on possession of similar indicators that had equal weightings. Low SES and multiple deprivation were characterized by the worst metrics of sixteen indicators of socio-economics and material deprivation in that locality. Other metrics such as access to electricity with a meter, living in a flat or bungalow with aluminum roofing sheet and water-closet, in a sparsely populated, low traffic density, planned neighborhood with tarred road and closed drainage; availability of pipe-borne water and use of gas for cooking characterized middle/affluent neighborhood.

The two studies from Nigeria and Uganda urban cities showed the clear links between various measures of SES, such as level of education and household income, household assets, and housing material, with household cooking and lighting fuel usage. Socio-economic status was also shown to influence the ability of households to engage in activities that can mitigate exposure to cooking fuel emissions. While these realizations are not new, the level of similarities regarding the types of fuels being used and what these fuels were used for among Africa’s urban poor was conspicuous. As just one example, Kerosene is commonly used for both household cooking and for lighting indoors in the urban context of SSA economically deprived populations. There are clear linkages between SES, household air pollution sources and exposure mitigating factors, and the common burdens in different regions of urban SSA. However, there remain limited, focused efforts to validate and harmonize measures of SES in the conduct of environmental epidemiological research in SSA that involve populations from urban slum environments, using tools and methods that demonstrates validity and reliability of the SES indicators. Being able to conceptualize and operationalize a harmonized measure of SES in urban SSA is vital not only for designing research studies but also for interpreting its results in defining the extent of exposure disease association and, more importantly, determining the size of health benefit achievable through feasible exposure reductions.

## 3. Improvements in Air Pollution Exposure Assessment in Resource Constrained Contexts

Significant disparities and limitations of air pollution exposure assessment approaches currently being implemented in different urban regions of Africa were highlighted at the symposium. The major disparity discussed is the lack of consistent access to air monitoring instruments in SSA countries to facilitate exposure assessment studies. Air monitors are typically obtained from international collaborators in the Global North and in most cases, borrowed for intermittent sampling periods, to support short-term research studies, and eventually returned at the conclusion of sampling campaigns. Such air pollution exposure assessments are thus heavily reliant on external partners—collaborations which may be fleeting due to funding constraints and funding imbalances globally. Short-term air sampling campaigns further limit the ability to accurately estimate chronic or acute exposures in a consistent and representative manner. While these disparities and limitations can partially be due to resource constraints, we note that many of these constraints are shared regionally and perceived as barriers to conducting high-quality exposure assessment for air pollution. These shared resource constraints can become an opportunity for enhanced collaboration and harmonization of data collection methods on the continent. Despite the relative lack of data on the diversified sources of household and ambient air pollution in SSA’s urban slum neighborhoods, these sources are likely to overlap across the region. It follows that the ways to mitigate household and ambient exposures will coincide as well, albeit with some inherent variability based on cultural differences, differences in the level economic development between regions, and differences in non-modifiable factors such as meteorology and geography. However, the shared characteristics can be harnessed to develop both survey-based and monitoring-based air pollution exposure assessment strategies that are standardized for SSA’s urban context. In addition, various low-cost air sensors and information and communication technologies (ICT) have emerged recently that can foster more data harmonization and data sharing. Freely available satellite remote sensing data can be fused with ground-level air monitoring for improvements in air pollution exposure assessment in SSA. Available satellite data products are varied and include (but not limited to) aerosol optical depth (AOD) available from Moderate Resolution Imaging Spectroradiometer (MODIS) and Visible Infrared Imaging Radiometer Suite (VIIRS) or precursor measurements of PM available from the TROPOspheric Monitoring Instrument (TROPOMI) and the Geostationary Operational Environmental Satellite Forward Processing (GEOS-FP). Such data products are able to provide high spatiotemporal resolution of pollutant measurements to help augment sparse air monitoring in SSA and support modeling efforts in the region [20,21,22,23]. Other satellite remote sensing data products also provide high spatial resolution of geographic features and climate variables important for air pollution modeling, such as digital elevation maps (DEM), land use characteristics (e.g., greenspace), and meteorological factors. Recently, application of novel machine learning (ML) methods using satellite data has shown promise in South Africa (ensemble averaging with AOD data) [22] and Ghana (deep learning with freely available satellite imagery data). Standardization of exposure assessment methods regionally and data science tools, which can integrate and analyze such data, should be promoted in the region to foster improvements in air pollution exposure assessment. Researchers may also leverage these emergent measurement, analysis, and ICTs tools for community engagement by sharing data with communities, citizen science, and crowdsourcing of environmental pollution data.

## 4. Harmonization in the Assessment of Health Outcomes and Biomonitoring of Contaminants

Global burden of disease estimates indicates that SSA suffers disproportionately from environmental pollution. However, the underlying data to support these estimates are not as reliable as data from higher-income countries [24]. There is a clear need to link and align SSA’s disease burden priority areas with environmental exposure data. However, the assessment of health outcomes and biomonitoring to assess contaminant exposures are very costly in cohort studies. Hence, the limited amount of data in resource-constraints areas such as SSA. This is another opportunity to pool resources on the continent. Harmonizing the assessment of health outcomes and biomonitoring of contaminants between studies and regions can facilitate pooling of epidemiologic data to promote more robust epidemiologic analyses in the region.

The benefits of cooperation and pooling of cohort data have already been enumerated by others [25]. These include, but are not limited to, improvements in the generalizability of epidemiological evidence, improvements in statistical power for small but important effects, identifying sources in heterogeneity of effects (which may be particularly important in the SSA context), maximizing existing resources, and cost efficiency [26]. For example, pooling cohort data in SSA could increase the statistical power to detect adverse effects from understudied yet potentially ubiquitous environmental chemical exposures in SSA, such as blood lead levels, chemical pesticides, plasticizers, and other endocrine disrupting compounds. Moreover, given the extensive research funding gaps facing SSA researchers, international research funding agencies are likely to find it appealing to fund such collaborative projects because of the benefits noted above. If environmental epidemiology researchers in Africa can harmonize methods at the outset, they can obviate the significant challenges [27] by pooling data post-hoc.

## 5. Reflections, Future Research Priorities and Conclusions

The Demographic and Health Surveys (DHS) Program has a measure of household air pollution (HAP) and water, sanitation, and hygiene (WASH) [28], which is applicable across all countries and has facilitated multi-country studies in the African region on the health effects of HAP and WASH with wide policy implications in the region. Such studies have informed the rolling out of clean energy solutions and development of WASH interventions in several African countries. Organizations such as the Africa Union Division of Health, Nutrition, and Population should coordinate a workshop for countries’ National Statistical Agencies and Ministries of Health to define and harmonize measures of SES across countries for operationalization in national surveys to enable data pooled from SSA countries for environmental epidemiological research to be comparable within and between countries. That way, such research will have wide policy relevance across the continent and help address the growing socioeconomic deprivation and deplorable environmental conditions in urban slums of Africa for population health gains.

Future research priorities in line with this commentary’s observations should focus on robust measurement of SES, the socioeconomic differentials in environmental pollution and human exposures in SSA regions to help understand how socioeconomic disadvantage can trigger environmental pollution as well as predisposes individuals to exposure. Such research is urgently needed to trigger sustained action for addressing socioeconomic inequalities in African countries.

In conclusion, we make a strong and compelling case for conceptualizing and harmonizing a measure of SES and environmental pollution in African countries to enable comparison of environmental epidemiological data across countries for the needed policy impact.

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
