# Peer review of "Harmonization of Epidemiologic Research Methods to Address the Environmental and Social Determinants of Urban Slum Health Challenges in Sub-Saharan Africa"

_ijerph, 2022, doi:10.3390/ijerph191811273_

Round 1
Reviewer 1 Report
Manuscript ID: ijerph-1794273
Title: Harmonization of Epidemiologic Research Methods to Address the Environmental and Social Determinants of Urban Slum Health Challenges in Sub-Saharan Africa
The authors of the reviewed commentary presented the finding of the recent environmental health studies from urban cities in East and West Africa, presented during a symposium sponsored by the Africa Chapter 13 of the International Society of Environmental Epidemiology (ISEE) in August 2020. The topic is significant and very well-articulated; however, I have the following concerns:
- The authors claim that this commentary outcome is to galvanize environmental epidemiologists in Sub-Saharan Africa (SSA) around shared data collection instruments and shared data platforms and facilitate the pooling of data across the continent. However, this statement needs more emphasis on the applicability of this suggestion. I would appreciate it if the authors could expand on this in improving air pollution exposure assessment in resource-constrained contexts.
- Add a Reflections and Future work section.
- Mention, under the Reflection section, how this work would filter through to government policy and a wider collective action.
Author Response
Thanks for your useful comments. The paper have been revised to reflect the comments. Please find the details in the attached document.

Reviewer 2 Report
See comments attached

Author Response
Thanks for your useful comments. They have been reflected in the revised manuscripts as much as possible. Please find the details in the attached,

Round 2
Reviewer 2 Report
I am satisfied with the way comments were addressed and I recommend this paper for publication.
The authors should just ensure that References are following the Journal Style in their final version for publication as there are lot of inconsistencies in this current version.